# Unveiling The Mask of Position-Information Pattern Through the Mist of Image Features

## Abstract

Recent studies show that paddings in convolutional neural networks encode absolute position information which can negatively affect the model performance for certain tasks. However, existing metrics for quantifying the strength of positional information remain unreliable and frequently lead to erroneous results. To address this issue, we propose novel metrics for measuring (and visualizing) the encoded positional information. We formally define the encoded information as PPP (Position-information Pattern from Padding) and conduct a series of experiments to study its properties as well as its formation. The proposed metrics measure the presence of positional information more reliably than the existing metrics based on PosENet and a test in F-Conv. We also demonstrate that for any extant (and proposed) padding schemes, PPP is primarily a learning artifact and is less dependent on the characteristics of the underlying padding schemes.

## 1   Introduction

Padding, one of the most fundamental components in neural network architectures, has received much less attention than other modules. Zero padding is frequently used in CNNs, perhaps due to its simplicity and low computational costs. This design preference remains almost unchanged in the past decade. Recent studies [1, 2, 3, 4] show that padding can implicitly provide a network model with positional information. Such positional information can cause unwanted side-effects by interfering and affecting other sources of position-sensitive cues (e.g., explicit coordinate inputs [5, 6, 7, 8, 9], embeddings [10], or boundary conditions of the model [4, 11, 12]). Furthermore, padding may lead to several unintended behaviors [5, 7, 8, 9], degrade model performance [10, 11, 12], or sometimes create blind spots [6]. Meanwhile, simply ignoring the padding pixels (known as no-padding or valid-padding) leads to the foveal effect [13, 14] that causes a model to become less attentive to the features on the image border. These observations motivate us to thoroughly investigate the phenomenon of positional encoding including the impact of commonly used padding schemes.

Conducting such a study requires a reliable metric to detect the presence of positional information introduced by padding, and more importantly, quantify its strength consistently. We observe that the existing methods for detecting and quantifying the strength of positional information yield inconsistent results. In Section 3, we revisit two closely related evaluation methods, PosENet [1] and F-Conv [3]. Our extensive experiments demonstrate that (a) metrics based on PosENet are unreliable with an unacceptably high variance, and (b) the 'Border Handling Variants' (BHV) test in F-Conv suffers from unaware confounding variables in its design, leading to unreliable test results.

---

The source codes and data collection scripts will be made publicly available.

Submitted to 36th Conference on Neural Information Processing Systems (NeurIPS 2022). Do not distribute.

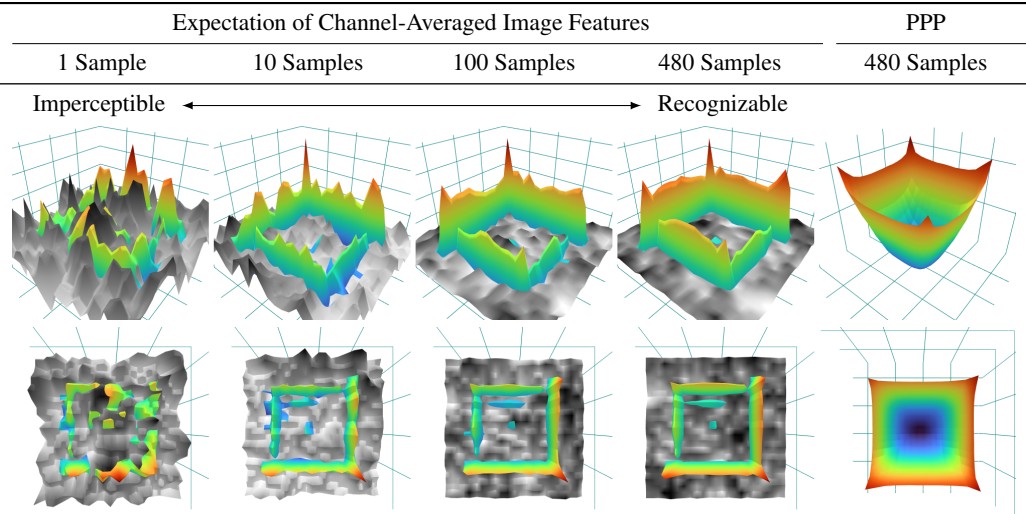

| Expectation of Channel-Averaged Image Features | | | | PPP |
|---|---|---|---|---|
| 1 Sample | 10 Samples | 100 Samples | 480 Samples | 480 Samples |

Imperceptible ← → Recognizable

Figure 1: **Position-information Pattern from Padding (PPP).** We propose a method that can consistently and effectively extract PPPs through the distributional difference between optimally-padded (gray-scale surfaces) and algorithmically-padded features (colored surfaces). The results show that the two distributions become distinguishable as the number of sample increases. Following the procedure in Section 2.2, we extract a clear view of PPP with the expectation of the pair-wise differences between optimally-padded and algorithmically-padded features. We render each visualization in tilted view (first row) and top view (second row). The colors represent the magnitude (blue/cold/weak to green/warm/strong) at each pixel. The features are extracted at the 3rd layer of interest (Appendix A) from a randn-padded (Section 2.4) ResNet50 pretrained on ImageNet.

In addition, we observe all commonly-used padding schemes actually encode consistent patterns underneath the highly dynamic model features. However, such a pattern is rather obscure, noisy, and visually imperceptible[1] in most cases. Fortunately, we show that such patterns can be consistently revealed with a sufficient number of samples by defining an optimal padding scheme (see Section 2.1 and Figure 1). We accordingly propose a new evaluation paradigm and develop a method to consistently detect the presence of the Position-information Pattern from Padding (PPP), which is a persistent pattern embedded in the model features to retain positional information. We present two metrics to measure the response of PPP from the signal-to-noise perspective and demonstrate its robustness and low deviation among different settings, each with multiple trials of training.

To weaken the effect of PPP, we design a padding scheme with built-in stochasticity to halt the model from constructing consistent patterns in Section 2.4. However, our experiments show that the models can still circumvent the stochasticity and end up consistently constructing certain PPPs. This observation suggests that a model likely constructs PPPs purposely to facilitate its training, rather than falsely or accidentally learning some filters that respond to padding features.

With reliable PPP metrics, we conduct a series of experiments to analyze the characteristics of PPP in Section 4.1. Specifically, we monitor the formation of PPP throughout each model training process in Section 4.3. The results show PPPs are formed expeditiously at the early stage of model training, slowly but steadily strengthened through time, and eventually shaped in clear and complete patterns. These results show that a model intentionally develops and reinforces PPPs to facilitate its learning process. Moreover, we observe the PPPs of all pretrained networks are significantly stronger than those in their initial states. This indicates an unbiased training procedure is of great importance in resolving the critical failures caused by PPP in numerous vision tasks [6, 7, 10, 11].

## 2 Observations and Methodology

In this section, we first define symbols for expressing the functionality of paddings and define the optimal-padding scheme. We then give a formal definition of Position-information Pattern from

---

[1]Except the zeros-padding is already well-known with its clear ring-shaped pattern [6, 1].

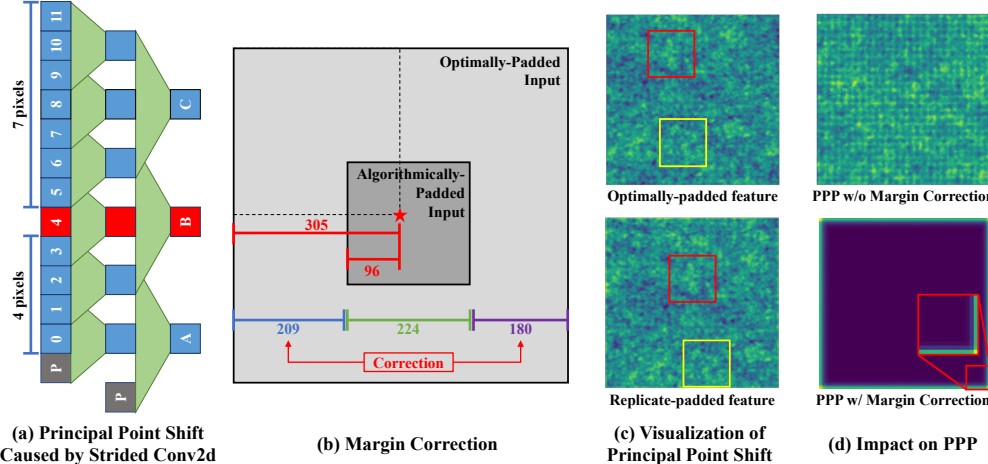

| (a) Principal Point Shift Caused by Strided Conv2d | (b) Margin Correction | (c) Visualization of Principal Point Shift | (d) Impact on PPP |

Figure 2: **Principal point shift.** (a) The stride-2 Conv2d only pads on one side, causing the principal point shift (red squares) in earlier layers. (b) Such a shift requires careful margin correction while aligning algorithmically-padded and optimally-padded features (we describe the details of point shift in Appendix A). (c) The shift is visible in the feature space (spade-shaped and question-mark-shaped patterns in the marked box). (d) It is crucial to correct the principal point shift while measuring PPP. The PPP calculation involves pixel-wise distance functions, which are not robust to spatial shifts [15].

Padding (PPP) and utilize the optimal-padding scheme to develop propose a method to capture PPP and measure its response with two metrics.

## 2.1 Optimal Padding

The process of capturing an image from the real world can be simplified as the 3D information of the environment is first projected onto an infinitely large 2D plane, and then the camera determines resolution as well as field-of-view to form an image from such infinitely large and continuous 2D signals [16, 17]. Let $S^* = \{s_n^*\}_{n=1}^N$ be a collection of such infinitely large and continuous 2D signals, and the collection of 2D images captured by cameras at a spatial size $(h_n, w_n)$ be $S' = \{s_n'\}_{n=1}^N$. A padding scheme produces a set of *algorithmically-padded* images $\hat{S} = \{\hat{s}_n\}_{n=1}^N$ by a padding function $\rho$:

$$\hat{s}_n[i,j] = \begin{cases} s_n'[i,j] = s^*[i,j] & \text{if } 0 < i < h_n \text{ and } 0 < j < w_n\,, \\ \rho(s_n', i, j) & \text{otherwise,} \end{cases} \quad (1)$$

where $i$ and $j$ are index of a pixel in the spatial dimension. We define a theoretical *optimally-padded* collection $S^\dagger = \{s_n^\dagger\}_{n=1}^N$ with an optimal-padding function $\rho^\dagger$ by:

$$s_n^\dagger[i,j] = \begin{cases} s_n'[i,j] & = s^*[i,j] & \text{if } 0 < i < h_n \text{ and } 0 < j < w_n\,, \\ \rho^\dagger(s_n', i, j) & = s^*[i,j] & \text{otherwise.} \end{cases} \quad (2)$$

In practice, such an *optimal*-padding scheme is difficult to achieve. However, it can be simulated if we have access to images beyond the sizes $(h_n, w_n)$ and artificially create $S'$.

## 2.2 Positional-information Pattern from Padding

As PPP has not been well defined in the literature, there is no effective metric to detect or quantify it. Ideally, PPP should have two properties. First, it is a spatial pattern as the padding pixels at different locations contribute differently to the formation of PPP. Its shape enables the network to develop and exploit the absolute positional information of each pixel, eventually leading to the unattended and undesirable effects in certain tasks [5, 6, 7, 8, 9, 10, 11].

Second, as it represents the positional information purely contributed by the padding, it is a constant term irrelevant to the image contents. Unfortunately, PPP shares space with image features, and these two spaces *interfere* with each other, causing the appearance of PPP extremely obscure in most cases (except zeros padding). Figure 1 shows if we visualize features sample-by-sample, there are no obvious differences between optimally-padded features (gray-scale surface) and algorithmically-

padded features (colored surface). Fortunately, if we assume the interferences between PPP and image features to be random, then its expectation over a large set of images will saturate to a constant bias and no longer hinder us from capturing PPP.

Based on these observations, we define PPP as the constant component independent of model inputs, and its presence is completely contributed by the existence of a padding scheme $\rho$. Given $\hat{S}$ and a model $F(\hat{s}; \theta, \rho)$, which $\theta$ is the model parameters and $\rho$ is a padding scheme applied to $F$. Let the model feature extracted at $k$-th layer be $f_{n,k} = F_k(\hat{s}_n; \theta, \rho)$, where $F_k$ is the model from the first layer to the $k$-th layer. The PPP at $k$-th layer ($PPP_k$) can be formulated by:

$$\text{PPP}_k = \mathbb{E}_n \left[ d \left( F_k(s_n^\dagger; \theta, \rho^\dagger), F_k(\hat{s}_n; \theta, \rho) \right) \right], \tag{3}$$

where $d(\cdot, \cdot)$ can be any distance function, and we use $\ell_1$ distance in this work.

**Pitfalls: feature misalignment.** It is important to note that, some CNN components can cause serious feature misalignment while computing PPP and leads to erroneous results. A typical example is *principal point shift*, where the uneven padding in stride-2 convolution causes the centers of features slightly drifted, as shown in Figure 2. Since the measurement of PPP requires perfect alignment, such a drift should be carefully considered while integrating PPP into new architectures. We further discuss the issue along with other pitfalls in Appendix A and provide three detailed examples of correcting the principal point shifting.

## 2.3 Metrics

In order to measure the strength of PPP, a proper baseline signal is needed. As discussed above, a strong PPP should be distinguishable from the interferences of the model features, so that the model can successfully extract the positional information from PPP. Thus, if we consider the model features as a background noise signal and PPP as the signal of interest, we can measure the significance of PPP using the signal-to-noise ratio (SNR). We define the SNR for PPP at $k$-th layer as:

$$\text{SNR-PPP}_k = \mu \left( \mathbb{E}_n \left[ || F_k(s_n^\dagger; \theta, \rho^\dagger) - F_k(\hat{s}_n; \theta, \rho) ||_1 \right] \right) / \sigma( F_k(\hat{s}_n; \theta, \rho) ), \tag{4}$$

where $\mu$ and $\sigma$ are the mean and standard deviation on the spatial dimensions.

However, SNR only measures the significance of the signal versus the noise but ignores the location of the signal. Given PPP is a spatially varying pattern, we further include Mean Absolute Error (MAE) to measure PPP versus the average of the noise map with:

$$\text{MAE-PPP}_k = \mathbb{E}_n \left[ \text{MAE} \left( F_k(s_n^\dagger; \theta, \rho^\dagger), F_k(\hat{s}_n; \theta, \rho) \right) \right]. \tag{5}$$

## 2.4 Randn Padding

Most of the existing padding schemes (e.g., zeros, reflect, replicate, circular) exhibit certain consistent patterns that can be easily detected by some designed convolutional kernels. One may argue that the nature of easy detectability can be a root cause of encouraging the models to learn to rely on these obvious patterns. This motivates us to design an additional sampling-based padding scheme without any consistent patterns, namely randn (i.e., random normal) padding, which produces dynamical values from a normal distribution while following the local statistics. We first determine the maximal and minimal values of a sliding window (which can be easily achieved with max-pooling), use the average of them as a proxy mean $\mu_p$, and use the difference between the mean and the maximal value as a proxy standard deviation $\sigma_p$. For each padding location, we sample the padding value according to a normal distribution $\mathcal{N}(\mu_p, \sigma_p^2)$ from the nearest sliding window. We include more implementation details in Appendix A.

Aside from creating a pattern-less padding scheme with sampling, the design of randn padding is based on several factors. The sampled padding pixels are allowed to occasionally exceed the min/max bound of the sliding window. Without breaking the min/max bound can introduce detectable patterns in certain extreme cases, such as a gradient-like feature that has its maximal intensity at the top-left corner and minimal intensity at the bottom-right corner. We also design the padding scheme to follow the local distribution. The padding exhibits a high entropy when the local variation is high, while degenerates to value repetition with imperceptible perturbations while padding a flat area. As

such, not only do the padding pixels exhibit less pattern, but it also prevents the padding pixels from breaking the features in the border region. We later show that a model still deliberately and incredibly built up PPP over time even with such a sophisticated padding scheme.

## 3 Revisiting Prior Work

In this section, we first reproduce two experiments from the prior art, which aim to assess positional information from paddings. We show several critical design issues in these experiments and discuss how these problems affect the drawn conclusions. Finally, we propose two additional experiments to quantify the amount of positional information embedded in the paddings.

### 3.1 PosENet

Islam *et al.* show zeros-padding provides CNN models positional information cues, and propose PosENet [1] to quantify the amount of positional information encoded within CNN features. A PosENet experiment involves several components: a pretrained CNN model $F$, a shallow CNN $E_{pem}$ (i.e., position encoding module), an image dataset $X = \{x_i\}_{i=1}^N$ to examine, and a constant target pattern $y$ (e.g., 2D Gaussian pattern). PosENet first extracts intermediate features at $k$-th layer with $f_{(i,k)} = F_k(x_i)$ using the pretrained CNN, and then optimizes $E_{pem}$ to minimize $\mathbb{E}_{i,k}[||E_{pem}(f_{(i,k)}) - y||_2]$. Finally, the amount of positional information is quantified by the average Spearman's correlation (SPC) and Mean Absolute Error (MAE) overall $E_{pem}(f_{(i,k)})$ toward $y$.

A critical issue with PosENet is the use of an optimization-based metric. It is sensitive to hyper-parameters with large variation. As shown in Table 2, for all the PosENet results, the standard deviation over five trials significantly dominates the differences between different types of paddings, and thus no definitive conclusions can be drawn. We also observed that PosENet can report NaN results in certain setups. Furthermore, PosENet quantifies the amount of positional information by the faithfulness of the final reconstruction. However, a better reconstruction does not have a clear relationship to *measuring* the strength and significance of positional information. For instance, the VGG architecture with zeros-padding in Table 2, PosENet cannot recognize the positional information has been strengthened after training, which can be seen in Figure 4. PosENet falsely assigns a much lower SPC to the fully pretrained model. Moreover, for the no-padding entries in Table 2, PosENet can still sometimes show responses to no-padding models, demonstrating it is a metric with an indefinite bias pending on the memorization ability of $E_{pem}$.

Another issue is that the no-padding scheme used in $E_{pem}$ is known to have the foveal effect [13, 14], where a model pays less attention to the information on the edge of inputs. Using such a padding scheme for detecting positional information from paddings, which is mostly concentrated on the edge of the feature maps, is less effective. This is an inevitable dilemma as PosENet aims to identify positional information from the padding of the pretrained $F$, while applying any padding scheme to $E_{pem}$ introduces intractable effects between the paddings of the two models.

### 3.2 F-Conv

Kayhan *et al.* propose a full-padding scheme (F-Conv) [3] and demonstrate it is more translational invariant than the alternatives. One of the critical results is on "border handling variants" (Exp 2 of [3]), which we call it BHV test. The BHV test creates a toy dataset, where each image has a black background with a green square and a red square in the foreground. The task is to predict if the red square is on the left of the green square (class 1), or vice versa (class 2). In addition, Kayhan *et al.* intentionally adds a *location bias* such that both squares are located in the upper half of the image for class 1, and located in the lower half of the image for class 2. During testing, a "similar test" inherits the same bias, while a "dissimilar test" exchanges the bias (i.e., both squares are in the lower half of the image for class 1). As a truly translation-invariant CNN model should not be affected by the location bias, it should focus on the relation between the red and green squares and perform similarly on both tests. Since the experimental results show that F-Conv performs best on the dissimilar test, it is concluded that F-Conv is less sensitive to the location bias. The authors also conclude the circular padding performs worse due to the behavior of wrapping the pixels to the other side of the image, which leads to confusion between two classes.

Table 1: **Background color as a critical confounding variable in BHV test.** We show that using a grey background similar to Figure 3 leads to discrepant results. The standard deviations are reported among 10 individual trials. We mark the best performance in green, and the worst two in red.

| Padding | F-Conv? | Black Background | | | | Grey Background | | | |
|---|---|---|---|---|---|---|---|---|---|
| | | Similar (%) | Dissimilar (%) | Diff (%) | Inconsistency (%) | Similar (%) | Dissimilar (%) | Diff (%) | Inconsistency (%) |
| Zeros | N | $99.83_{\pm 0.00}$ | $3.21_{\pm 8.35}$ | $-87.68$ | $95.81_{\pm 2.07}$ | $100.00_{\pm 0.00}$ | $4.96_{\pm 5.93}$ | $-95.04$ | $97.85_{\pm 4.55}$ |
| | Y | $89.24_{\pm 0.98}$ | $89.24_{\pm 0.98}$ | $0.00$ | $18.02_{\pm 8.08}$ | $100.00_{\pm 0.00}$ | $4.77_{\pm 6.52}$ | $-95.23$ | $96.79_{\pm 7.13}$ |
| Circular | N | $80.31_{\pm 3.23}$ | $80.31_{\pm 3.23}$ | $0.00$ | $34.25_{\pm 8.32}$ | $72.75_{\pm 0.96}$ | $72.75_{\pm 0.96}$ | $0.00$ | $26.30_{\pm 5.55}$ |
| | Y | $99.20_{\pm 0.23}$ | $93.14_{\pm 2.88}$ | $-6.06$ | $18.48_{\pm 3.55}$ | $98.26_{\pm 0.50}$ | $92.40_{\pm 4.23}$ | $-5.87$ | $28.67_{\pm 6.18}$ |
| Reflect | N | $100.00_{\pm 0.00}$ | $15.67_{\pm 12.72}$ | $-84.33$ | $91.18_{\pm 13.19}$ | $100.00_{\pm 0.00}$ | $19.96_{\pm 13.54}$ | $-80.04$ | $90.33_{\pm 11.95}$ |
| | Y | $100.00_{\pm 0.00}$ | $11.70_{\pm 15.38}$ | $-88.30$ | $97.33_{\pm 6.16}$ | $100.00_{\pm 0.00}$ | $17.16_{\pm 12.19}$ | $-82.84$ | $98.13_{\pm 3.44}$ |
| Replicate | N | $100.00_{\pm 0.00}$ | $43.39_{\pm 11.42}$ | $-56.61$ | $75.32_{\pm 8.20}$ | $100.00_{\pm 0.00}$ | $33.16_{\pm 6.42}$ | $-66.83$ | $84.09_{\pm 6.47}$ |
| | Y | $98.32_{\pm 0.39}$ | $93.65_{\pm 1.36}$ | $-4.67$ | $32.60_{\pm 4.97}$ | $97.17_{\pm 0.48}$ | $94.99_{\pm 1.20}$ | $-2.18$ | $32.15_{\pm 5.11}$ |
| Randn | N | $100.00_{\pm 0.00}$ | $10.31_{\pm 12.56}$ | $-89.70$ | $94.88_{\pm 5.55}$ | $99.97_{\pm 0.13}$ | $35.47_{\pm 10.82}$ | $-64.50$ | $83.59_{\pm 8.48}$ |
| | Y | $100.00_{\pm 0.00}$ | $20.80_{\pm 14.15}$ | $-79.20$ | $92.54_{\pm 8.37}$ | $77.28_{\pm 16.13}$ | $66.70_{\pm 11.58}$ | $-10.59$ | $45.70_{\pm 20.62}$ |
| No-pad | - | $100.00_{\pm 0.00}$ | $3.21_{\pm 8.35}$ | $-96.79$ | $95.81_{\pm 2.07}$ | $100.00_{\pm 0.00}$ | $30.07_{\pm 4.06}$ | $-69.93$ | $81.30_{\pm 2.44}$ |

However, as shown in Figure 3, we find the experimental design does not consider a crucial confounding variable: the black background has a zero intensity, making zeros padding the optimal padding that perfectly follows the background distribution. In Table 1, we show that the dissimilar test is no longer in favor of F-Conv zeros after changing the background color to grey. We also show that F-Conv replicate and F-Conv circular perform best on the dissimilar test, which is different from the original observation.

Finally, we report an additional inconsistency rate to show that the CNN architecture used in the BHV test actually has access to the absolute position of the squares. Given a random sample in class 1, we create a *trajectory* of samples by simultaneously moving the two squares to the bottom of the canvas and recording the CNN-model prediction in all intermediate states. We label a trajectory to be *inconsistent* if the prediction of the CNN-model switches classes at any step of the trajectory. A CNN model with no access to the absolute-position information should have all trajectories maintaining consistent predictions, with $0\%$ inconsistency. Table 1

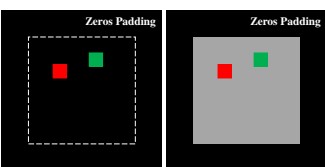

Figure 3: The BHV test trains a binary classifier to predict the relative position of the two colored squares. It hypothesizes if the padding provides no positional information, the classifier will only focus on the relative position of the two squares. (Left) The black background is a confounding variable. (Right) Zeros padding no-longer pads optimum values after changing the background color.

shows the inconsistent ratio over 228 uniformly sampled trajectories, where all models maintain high inconsistency rates, even with a no-padding architecture. These results show that the CNN model used in the BHV test is not translation invariant. This can be attributed to that a CNN model has a large receptive field covering the whole experiment canvas, therefore capable of gradually constructing absolute coordinates for each input pixel. Note that we only show the design of the BHV test is not suitable for quantifying the amount of positional information exhibited in a CNN model. Such a conclusion does not imply that F-Conv cannot potentially improve the translation-invariant property of CNNs.

## 4 Experiments and Analysis

**Datasets** Since most vision models are trained on tasks for recognizing objects, an image collection containing a diverse object appearance is more suitable for the task. We collect a set of $480$ satellite images at $2{,}048 \times 2{,}048$ pixels from Google Map for experiments. All the PPP metrics are measured with this image collection. We crop such images depending on the requested input image sizes and principal point shifts from each model (see Appendix A for details). We will release the script for collecting and composing these large images.

### 4.1 Visualizing Position-information Pattern from Padding (PPP)

We start with visualizing PPP in Figure 4. All the visualizations are conducted at the 4th layer of interest as detailed in Appendix A. We compute PPP using Eq. 3 and $\ell_1$ norm as the distance metric,

Table 2: **Comparing PosENet and our proposed PPP metrics.** The standard deviation is computed by five different pretrained models for each test. The performance shows the accuracy for the classification task or weighted F-measure score [18] for the saliency object detection task. Note that we use 2D Gaussian as PosENet reconstruction pattern, and the PPP metrics are measured at the 4th layer of interest. Here, ($^*$) indicates a NaN is reported in any of the trials, and ($\uparrow$) indicates a higher value corresponds to stronger positional information or better performance on the task (vice versa for ($\downarrow$)). For each group of pretrained models, we label the strongest and weakest positional information response with red and blue.

| Model | Padding | Pretrained | PosENet SPC ($\uparrow$) | PosENet MAE ($\downarrow$) | PPP (ours) SNR-PPP ($\uparrow$) | PPP (ours) MAE-PPP ($\uparrow$) | Performance ($\uparrow$) |
|---|---|---|---|---|---|---|---|
| VGG-19 | Zeros | $\times$ | $0.518_{\pm0.121}$ | $0.184_{\pm0.004}$ | $0.0665_{\pm0.0024}$ | $0.0132_{\pm0.0006}$ | - |
| | | ImageNet | $0.142_{\pm0.139}$ | $0.194_{\pm0.006}$ | $1.2289_{\pm0.0613}$ | $0.0176_{\pm0.0005}$ | $74.0972_{\pm0.0870}$ |
| | Circular | $\times$ | $0.001_{\pm0.092}$ | $0.197_{\pm0.002}$ | $0.0000_{\pm0.0000}$ | $0.0000_{\pm0.0000}$ | - |
| | | ImageNet | $0.102_{\pm0.136}$ | $0.197_{\pm0.007}$ | $1.1488_{\pm0.0589}$ | $0.0158_{\pm0.0006}$ | $74.4716_{\pm0.0863}$ |
| | Reflect | $\times$ | $0.001_{\pm0.091}$ | $0.197_{\pm0.002}$ | $0.0000_{\pm0.0000}$ | $0.0000_{\pm0.0000}$ | - |
| | | ImageNet | $0.116_{\pm0.134}$ | $0.195_{\pm0.006}$ | $1.2022_{\pm0.0226}$ | $0.0158_{\pm0.0002}$ | $74.0516_{\pm0.0621}$ |
| | Replicate | $\times$ | $0.001_{\pm0.091}$ | $0.197_{\pm0.002}$ | $0.0000_{\pm0.0000}$ | $0.0000_{\pm0.0000}$ | - |
| | | ImageNet | $0.116_{\pm0.132}$ | $0.195_{\pm0.006}$ | $1.2494_{\pm0.0258}$ | $0.0144_{\pm0.0009}$ | $73.9964_{\pm0.1079}$ |
| | Randn | $\times$ | $0.001_{\pm0.093}$ | $0.197_{\pm0.002}$ | $0.0000_{\pm0.0000}$ | $0.0000_{\pm0.0000}$ | - |
| | | ImageNet | $0.115_{\pm0.146}$ | $0.195_{\pm0.006}$ | $1.2366_{\pm0.0774}$ | $0.0182_{\pm0.0012}$ | $73.7716_{\pm0.0758}$ |
| | No-padding | $\times$ | $0.000_{\pm0.091}$ | $0.197_{\pm0.002}$ | $0.0000_{\pm0.0000}$ | $0.0000_{\pm0.0000}$ | - |
| | | ImageNet | $0.001_{\pm0.220}$ | $0.203_{\pm0.012}$ | $0.0000_{\pm0.0000}$ | $0.0000_{\pm0.0000}$ | $62.0396_{\pm0.0830}$ |
| VGG16-SOD | Zeros | $\times$ | $0.682_{\pm0.099}$ | $0.171_{\pm0.008}$ | $0.0306_{\pm0.0020}$ | $0.0068_{\pm0.0007}$ | - |
| | | DUTS | $0.343_{\pm0.151}$ | $0.186_{\pm0.011}$ | $0.2429_{\pm0.0035}$ | $0.0049_{\pm0.0001}$ | $0.6269_{\pm0.0015}$ |
| | Circular | $\times$ | $0.001_{\pm0.081}$ | $0.197_{\pm0.002}$ | $0.0000_{\pm0.0000}$ | $0.0000_{\pm0.0000}$ | - |
| | | DUTS | $0.158_{\pm0.188}$ | $0.196_{\pm0.013}$ | $0.2677_{\pm0.0062}$ | $0.0062_{\pm0.0001}$ | $0.6260_{\pm0.0009}$ |
| | Reflect | $\times$ | $-0.002_{\pm0.080}$ | $0.197_{\pm0.002}$ | $0.0000_{\pm0.0000}$ | $0.0000_{\pm0.0000}$ | - |
| | | DUTS | $0.160_{\pm0.223}$ | $0.195_{\pm0.014}$ | $0.1972_{\pm0.0024}$ | $0.0053_{\pm0.0001}$ | $0.6243_{\pm0.0022}$ |
| | Replicate | $\times$ | $-0.002_{\pm0.087}$ | $0.197_{\pm0.002}$ | $0.0000_{\pm0.0000}$ | $0.0000_{\pm0.0000}$ | - |
| | | DUTS | $0.075_{\pm0.174}$ | $0.201_{\pm0.010}$ | $0.1908_{\pm0.0056}$ | $0.0043_{\pm0.0002}$ | $0.6255_{\pm0.0013}$ |
| | Randn | $\times$ | $0.000_{\pm0.082}$ | $0.197_{\pm0.002}$ | $0.0000_{\pm0.0000}$ | $0.0000_{\pm0.0000}$ | - |
| | | DUTS | $0.004_{\pm0.106}$ | $0.196_{\pm0.001}$ | $0.0005_{\pm0.0001}$ | $0.0001_{\pm0.0000}$ | $0.2570_{\pm0.0022}$ |
| | No-padding | $\times$ | $0.000_{\pm0.087}$ | $0.197_{\pm0.002}$ | $0.0000_{\pm0.0000}$ | $0.0000_{\pm0.0000}$ | - |
| | | DUTS | $0.003_{\pm0.252}$ | $0.200_{\pm0.010}$ | $0.0000_{\pm0.0000}$ | $0.0000_{\pm0.0000}$ | $0.4759_{\pm0.0013}$ |
| ResNet50 | Zeros | $\times$ | $0.096_{\pm0.118}$ | $0.196_{\pm0.003}$ | $0.0918_{\pm0.0119}$ | $0.0052_{\pm0.0004}$ | - |
| | | ImageNet | $0.329_{\pm0.201}$ | $0.185_{\pm0.011}$ | $0.8171_{\pm0.0173}$ | $0.0162_{\pm0.0012}$ | $75.6856_{\pm0.0924}$ |
| | Circular | $\times$ | $^*0.027_{\pm0.093}$ | $^*0.197_{\pm0.003}$ | $0.0454_{\pm0.0041}$ | $0.0032_{\pm0.0004}$ | - |
| | | ImageNet | $0.184_{\pm0.201}$ | $0.192_{\pm0.010}$ | $0.7018_{\pm0.0320}$ | $0.0188_{\pm0.0016}$ | $76.1432_{\pm0.1026}$ |
| | Reflect | $\times$ | $^*0.004_{\pm0.094}$ | $^*0.198_{\pm0.003}$ | $0.0291_{\pm0.0017}$ | $0.0018_{\pm0.0001}$ | - |
| | | ImageNet | $0.293_{\pm0.181}$ | $0.187_{\pm0.009}$ | $0.6960_{\pm0.0221}$ | $0.0150_{\pm0.0004}$ | $75.5068_{\pm0.1213}$ |
| | Replicate | $\times$ | $^*0.002_{\pm0.094}$ | $^*0.198_{\pm0.003}$ | $0.0226_{\pm0.0013}$ | $0.0015_{\pm0.0001}$ | - |
| | | ImageNet | $0.347_{\pm0.205}$ | $0.184_{\pm0.012}$ | $0.7461_{\pm0.0254}$ | $0.0138_{\pm0.0003}$ | $75.6122_{\pm0.0911}$ |
| | Randn | $\times$ | $^*0.006_{\pm0.090}$ | $^*0.198_{\pm0.003}$ | $0.0326_{\pm0.0016}$ | $0.0020_{\pm0.0002}$ | - |
| | | ImageNet | $0.358_{\pm0.240}$ | $0.181_{\pm0.016}$ | $0.6648_{\pm0.0204}$ | $0.0147_{\pm0.0007}$ | $75.3076_{\pm0.1016}$ |
| EfficientNet | Zeros | $\times$ | $0.360_{\pm0.327}$ | $0.180_{\pm0.026}$ | $0.5074_{\pm0.0260}$ | $0.0398_{\pm0.0027}$ | - |
| | | ImageNet | $0.667_{\pm0.111}$ | $0.166_{\pm0.014}$ | $0.7590_{\pm0.0208}$ | $0.0471_{\pm0.0022}$ | $61.8652_{\pm0.1380}$ |
| | Circular | $\times$ | $0.004_{\pm0.192}$ | $0.205_{\pm0.013}$ | $0.3008_{\pm0.0883}$ | $0.0222_{\pm0.0048}$ | - |
| | | ImageNet | $0.020_{\pm0.123}$ | $0.203_{\pm0.009}$ | $0.4326_{\pm0.0251}$ | $0.0256_{\pm0.0017}$ | $61.2208_{\pm0.2128}$ |
| | Reflect | $\times$ | $0.003_{\pm0.175}$ | $0.205_{\pm0.012}$ | $0.2245_{\pm0.0639}$ | $0.0183_{\pm0.0053}$ | - |
| | | ImageNet | $0.062_{\pm0.116}$ | $0.201_{\pm0.008}$ | $0.4667_{\pm0.0232}$ | $0.0268_{\pm0.0014}$ | $60.4164_{\pm0.2924}$ |
| | Replicate | $\times$ | $0.004_{\pm0.183}$ | $0.205_{\pm0.013}$ | $0.2634_{\pm0.0748}$ | $0.0206_{\pm0.0035}$ | - |
| | | ImageNet | $0.131_{\pm0.139}$ | $0.197_{\pm0.008}$ | $0.5257_{\pm0.0334}$ | $0.0279_{\pm0.0007}$ | $60.9804_{\pm0.2134}$ |
| | Randn | $\times$ | $0.001_{\pm0.190}$ | $0.202_{\pm0.011}$ | $0.3606_{\pm0.0505}$ | $0.0248_{\pm0.0031}$ | - |
| | | ImageNet | $0.324_{\pm0.210}$ | $0.189_{\pm0.012}$ | $0.5686_{\pm0.0112}$ | $0.0209_{\pm0.0011}$ | $58.6392_{\pm0.2739}$ |

then average the resulting PPP in the channel dimension to generate a gray-scale image. Since the quantities are small and difficult to perceive, we normalize the gray-scale image to $[0, 1]$ range, and thus the colors between images are not directly comparable.

In all scenarios, a noticeable difference is that PPP spreads out after pretraining on ImageNet. In Table 2, the PPP-SNR of the VGG19 and ResNet50 also reflects that the response of PPP is significantly strengthened after model training. That is, the model training has substantial effects on the construction of PPP. Although the formation of padding pattern is suggested to mainly caused by

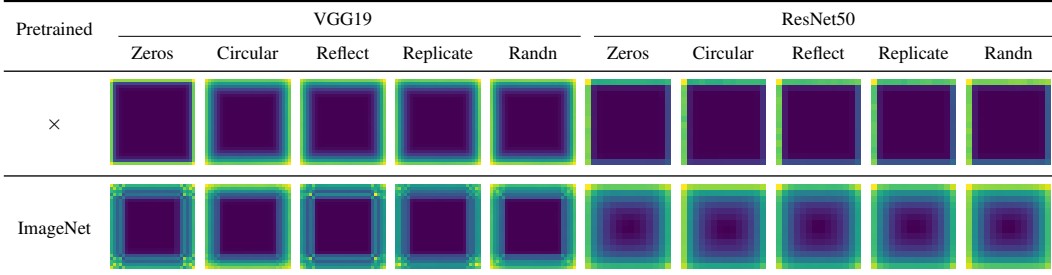

| Pretrained | VGG19 | | | | | ResNet50 | | | | |
| --- | --- | --- | --- | --- | --- | --- | --- | --- | --- | --- |
| | Zeros | Circular | Reflect | Replicate | Randn | Zeros | Circular | Reflect | Replicate | Randn |
| × | | | | | | | | | | |
| ImageNet | | | | | | | | | | |

Figure 4: **Visualization of Position-Information Pattern from Padding (PPP).** The visualizations are calculated based on Eq. 3 over 480 GMap samples extracted at the 3rd layer-of-interest (Appendix A). The results show that the pretrained model significantly reinforces PPP compared to randomly initialized networks. Note that each image is normalized to $[0, 1]$ separately, therefore the colors between images are not comparable. More visualizations are presented in Appendix B.

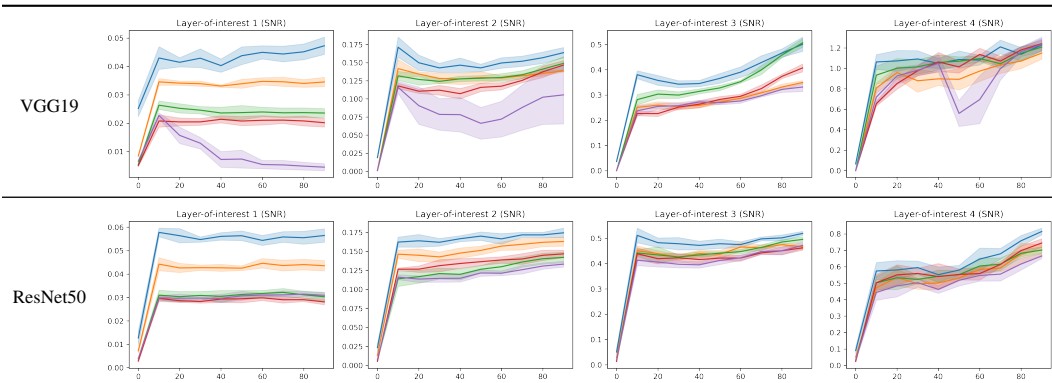

Figure 5: **Chronological PPP.** We quantify PPP every 10 epochs and plot its development in four different layer of depth (the rightmost layer is the one closest to model output). All curves consistently show a sudden surge at the early stage, and all the later layers are slowly but steadily gaining stronger PPP until the end of training. The shadow region represents standard deviations among 5 individual training episodes. The colors represent zeros, circular, reflect, replicate, and randn paddings.

the distributional difference between features and paddings [6], our results show that it only increases the response slightly, compared to the considerable PPP-SNR gain through training.

Another intriguing observation is that, despite some variations in the detailed patterns, the overall structure of PPP remains similar. Regardless of padding minimum values with zero-padding (consider the features are processed with ReLU activation), randn-padding that can sometimes produce large quantities by chance, or the unbalanced initial state of ResNet50 caused by strided convolution (the first row of ResNet50 in Figure 4), all models tend to have the maximal PPP response in the corner of the features after fully trained. While the underlying mechanism causing such consistent preferences remains unknown, such preferences may be an important factor to consider in future model design.

### 4.2 Quantifying PPP and Comparing with PosENet

Table 2 shows the measurements of PPP and PosENet on various architectures and padding schemes. We train five models for each setup and measure the standard deviation of these models. Our PPP metrics have significantly lower standard deviations compared to PosENet, where the standard deviation dominates the differences between padding variants, and thus the quantities from PosENet cannot provide sufficient information for any analysis. The main reason that PosENet has such a large variation is due to its optimization-based formulation, and thus the final quantities highly depend on the convergence of the PosENet training. In fact, we also observe a similar level of standard deviation even when the PosENet is measured on the same model for multiple trials. On the other hand, PPP metrics are based on a closed-form formulation, and thus the variations are only introduced by the differences among the parameters of the pretrained models. Furthermore, PosENet frequently reports

positive SPC responses from no-padding models, as shown in its large standard deviation. In contrast, PPP has zero response to no-padding models by definition, and therefore is less biased for measuring the positional information from padding.

SNR-PPP and MAE-PPP assess the response of PPP from two different perspectives, the ratio of the overall PPP magnitude to the image feature variation, and the position-aware average gain of PPP. Despite both measuring the PPP gain and mostly following similar trends, the two metrics can sometimes have discrepancies, such as the randn padding case in EfficientNet pretrained on ImageNet in Table 2. We note that the two metrics should be both measured and considered altogether.

Although certain paddings seem to have lower SNR-PPP or MAE-PPP on trained networks, we find the differences are not significant when comparing the extremely low SNR-PPP and MAE-PPP from the randomly initialized networks. In most cases, the network can effectively construct its PPP, even with the highly stochastic randn padding. The only exception seems to be the case of randn padding in the salient object detection (SOD) task, where the network fails to achieve a compatible performance to other paddings[2]. The results show that the model training plays an important role in the formation of PPP, and perhaps its contribution is much larger than which underlying padding scheme is being used. This motivates us to further analyze the PPP formulation during model training.

### 4.3 Chronological PPP

To understand the formulation of PPP through time, we snapshot checkpoints every 10 epochs for all training episodes. By measuring the PPP metrics at all the checkpoints, we plot a chronological curve and monitor the progress of PPP. We train 5 individual models for each pair of model-padding setup and report the standard deviations, which demonstrates the significance of the trend.

Figure 5 shows all models achieve a significant gain of PPP within the first 10 epochs in all intermediate layers. Most models continuously increase their PPP as training proceeds, especially in the fourth layer of interest, which is the last output from the convolutional layers before the final linear projection. Another interesting observation is that our randn padding, which is designed to be less easily detectable with built-in stochasticity, indeed shows less PPP built-up at the intermediate stages in certain layers. However, the network still adjusts the behavior and ends up forming complete PPPs at the fourth layer of interest in all scenarios. All these evidences show that the network builds PPP purposely as a favorable representation to assist its learning.

## 5 Conclusion and Limitations

In this paper, we develop a reliable method for measuring PPP and conduct a series of analyses toward understanding the formation and properties of PPP. Through a large-scale study, we demonstrate that PPP is a representation that the network favorably develops as a part of its learning process, and its formation has weak connections to the underlying padding algorithm. We show that reliable PPP metrics are important steps for understanding the effects of PPPs in different tasks, and useful for measuring the effectiveness of future methods in debiasing PPP.

However, an unfortunate and inevitable limitation of the PPP metrics is that their measure is biased by the model architecture and parameters. Since the PPP metrics are based on the distributional differences between the paired model outputs (i.e., optimal padding to algorithmic padding), different architecture and layers of depth exhibit different and intractable biases due to different interactions between PPP and model parameters. Such a bias makes PPP metrics less useful for evaluating models, and therefore cannot be used to study the effect of architectural changes. This limitation is inevitable for any (and all existing) metric that attempts to measure PPP using the outputs of a model. We note future studies in measuring PPP without model inferences[3] will be an important step toward tackling and understanding the property of PPP under different architectural choices.

---

[2]We follow PosENet that evaluates PiCANet [19] on the SOD task. PiCANet is initialized by a model pretrained on ImageNet (with zero padding). The discrepancy in the padding scheme can be the major cause of failure while training the network on SOD task with randn padding.

[3]A related analogy of the contradictory problem can be found in neural architecture search literature [20].

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
