# Unveiling The Mask of Position-Information Pattern Through the Mist of Image Features

## A    Implementation Details

### A.1    Architecture and Feature Alignments

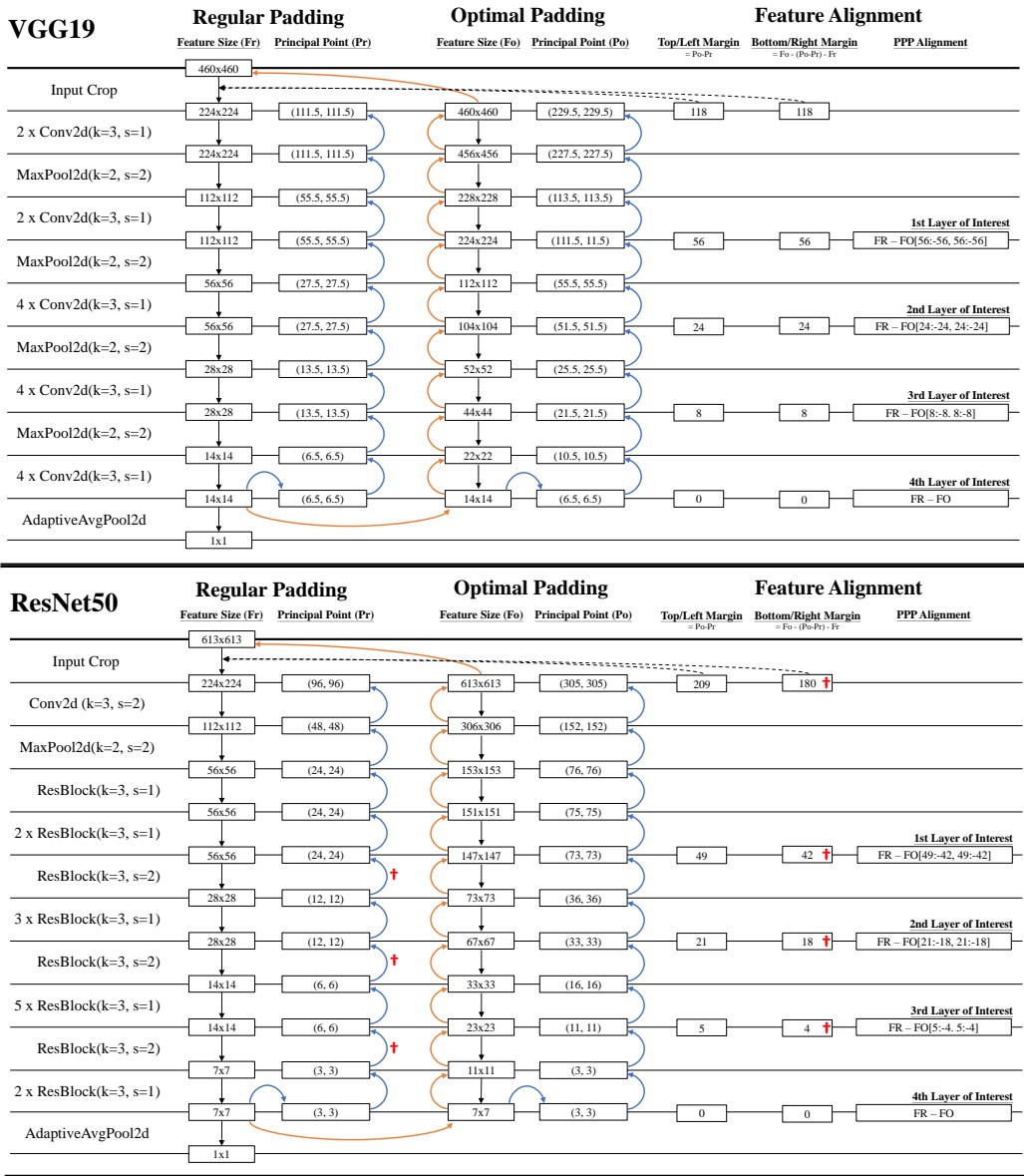

Figure 1: **The architecture for VGG19 and ResNet50 used in the paper.** We mark the calculation of optimal padding in orange arrows and principal point in blue arrows. We label the layers of interest that are used in the paper. The red † indicates where a principal point shift is identified.

Submitted to 36th Conference on Neural Information Processing Systems (NeurIPS 2022). Do not distribute.

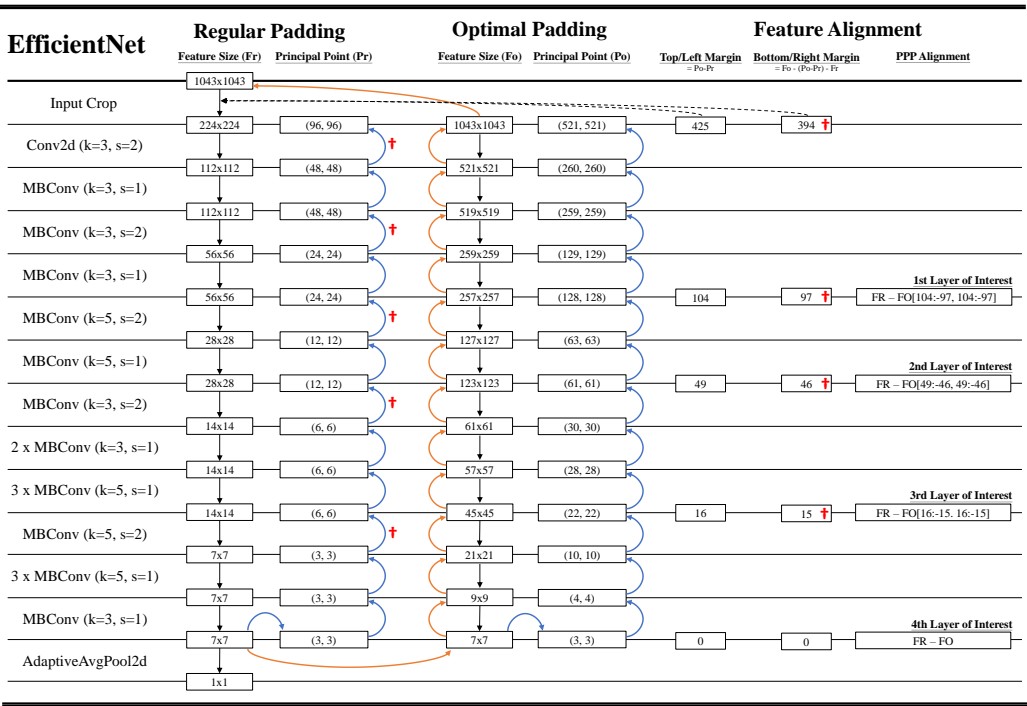

Figure 2: **The architecture for EfficientNet used in the paper.** We mark the calculation of optimal padding in orange arrows and principal point in blue arrows. We label the layers of interest that are used in the paper. The red † indicates where a principal point shift is identified.

## A.2  PPP Feature Misalignment

There are several pitfalls in visualizing and quantifying PPP. We identify two critical pitfalls from the architectures we implemented. However, these may not be sufficient to cover all potential issues while integrated into other architectures. Therefore one must be alerted to any unusual behavior (e.g., Figure 2(d) in the main paper) throughout their implementation.

**Principal point shifting.** Conv2d has a hidden behavior that few people are aware of, the operation is one-pixel skewed while applying a stride-two Conv2d on even-shaped features. To understand how does the one-pixel shift happen, we first define the principal point of a feature map. We first define the principal point of the last feature map as the center pixel (note that we define it as the middle-point between the center-two pixels in case the last feature size is even). Then, we recursively define the principal point of the $(N-1)$-th layer as the pixel that positions at the center of the Conv2d receptive field that mainly forms the principal point of the $N$-th layer. In the case of optimally-padded features, the principal points in every layer are the center of the feature map. But, as shown in Figure 2(a), the principal point of algorithmically-padded features will have a one-pixel shift when a stride-2 convolution is applied to even-shaped features, which can be further amplified as more layers stack up. Such a skew causes the principal points of algorithmically-padded features shift several pixels away from the principal points of optimally-padded features. As PPP metrics use pixel-wise subtraction to distinguish the image content from PPP, the misalignment becomes a critical issue, since the image contents are no longer aligned and subtractable.

In Figure 1 and Figure 2, we show the procedure of calculating the principal point in blue arrows and marking the values impacted by principal point shift with red †. For the ResNet50 architecture, the principal point shift accumulates to $16(= 224/2 - 96)$ pixels in the early layers.

Fortunately, such a displacement can be fixed by adding corrections to how we calculate the feature margins. As shown in Figure 2(b), the concept of the margin correction is to make the two principal

27  points overlapping each other after adding the margin. In the example, the left-right margins are
28  corrected to $(209, 180)$ (instead of the more intuitive choice of $(195, 194)$ or $(194.5, 194.6)$).

29  We also show how the principal point shift visually looking like in Figure 2(c), notice the patterns
30  have right-bottom shifted 16 pixels. As shown in Figure 2(d), failing to identify the principal point
31  shift will result in checkerboard artifacts while calculating PPP , and adding correction eliminates the
32  artifacts.

33  **Maxpooling misalignment.** This is a hypothetical condition that may potentially happen but has not
34  been observed in the three architectures we tested. Consider a case of a Maxpooling layer of window
35  size 2 and stride 2, the sliding windows of each pooling operation have no overlap, therefore the
36  initial index of the first sliding window solely determines the spatial location of all sliding windows.
37  Accordingly, there is a chance that the initial condition of the optimally-padded features causes all of
38  its sliding windows are 1-pixel misaligned to the algorithmically-padded features. Fortunately, the
39  condition can be easily determined by calculating the top and left margin of the feature alignment
40  (similar to the aforementioned principal point shift calculation). For the case of a Maxpooling layer
41  of window size 2 and stride 2, the misalignment will not happen if the top and left margins are even
42  numbers, and that is exactly the case for both VGG19 and ResNet50, as shown in Figure **??**.

43  ## A.3   Randn Padding

44  A critical implementation detail is that such a padding scheme must be applied before activation func-
45  tions. Since the paddings are based on the distribution within sliding windows, activation functions
46  such as ReLU, which clamps all negative values, can discard a significant amount of information be-
47  forehand. Instead of the traditional use of padding-convolution-normalization-activation, we modify
48  the order to convolution-normalization-padding-activation. Note that such a change of order does not
49  affect the behavior or results of other padding schemes.

50  ## A.4   Acknowledging Open-Source Contributors

51  Our implementation reuses codes from several open-source codebases, which greatly supports our
52  development. The repositories used in the paper are F-Conv [1], torchvision [2] and Pytorch-cifar [3].

## B  More PPP Visualizations

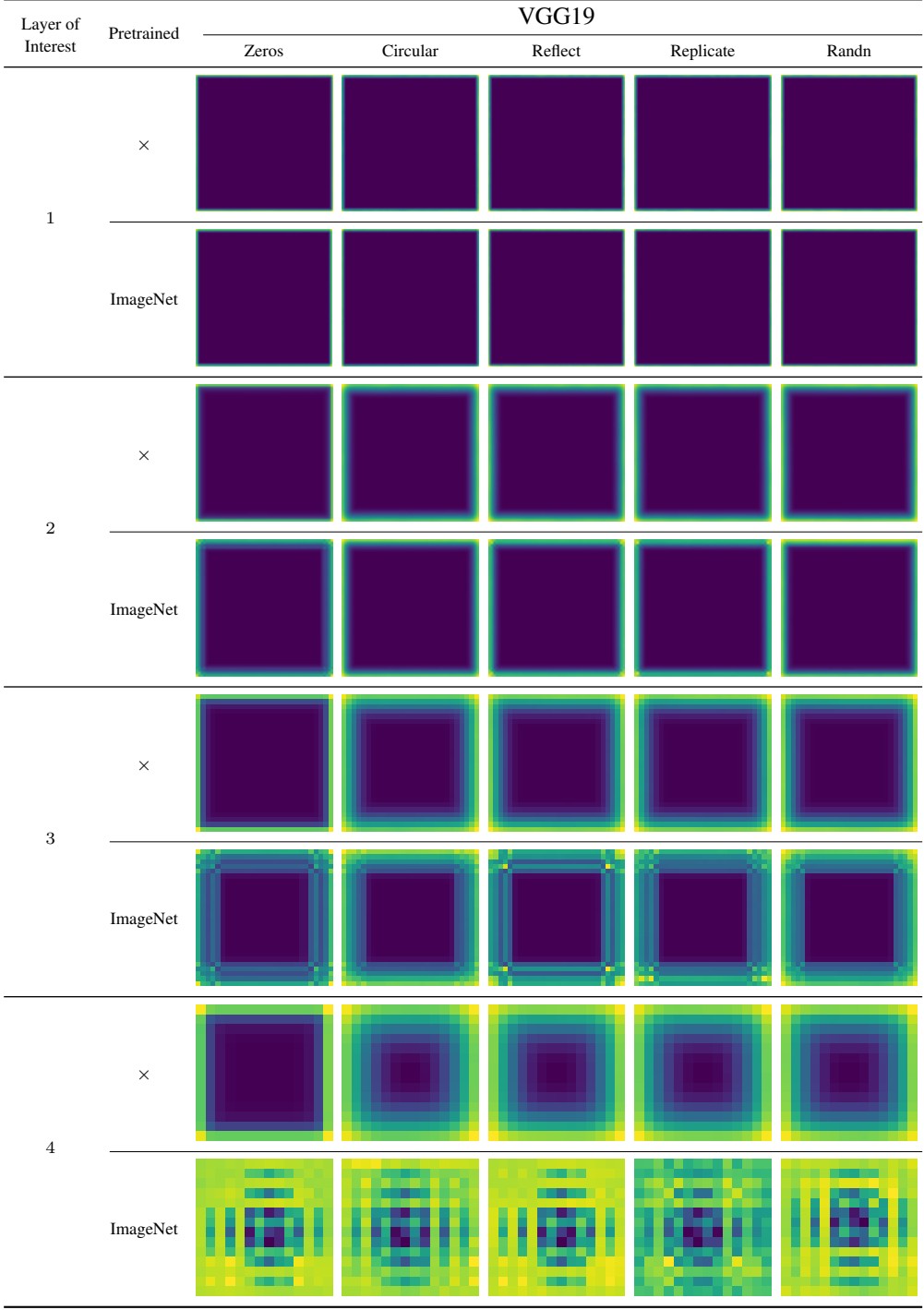

Figure 3: **Visualization of Position-Information Pattern from Padding (PPP).** The visualizations are calculated based on Eq. 3 over $480$ GMap samples. The results show that the pretrained model significantly reinforces PPP compared to randomly initialized networks. Note that each image is normalized to $[0, 1]$ separately, therefore the colors between images are not comparable.

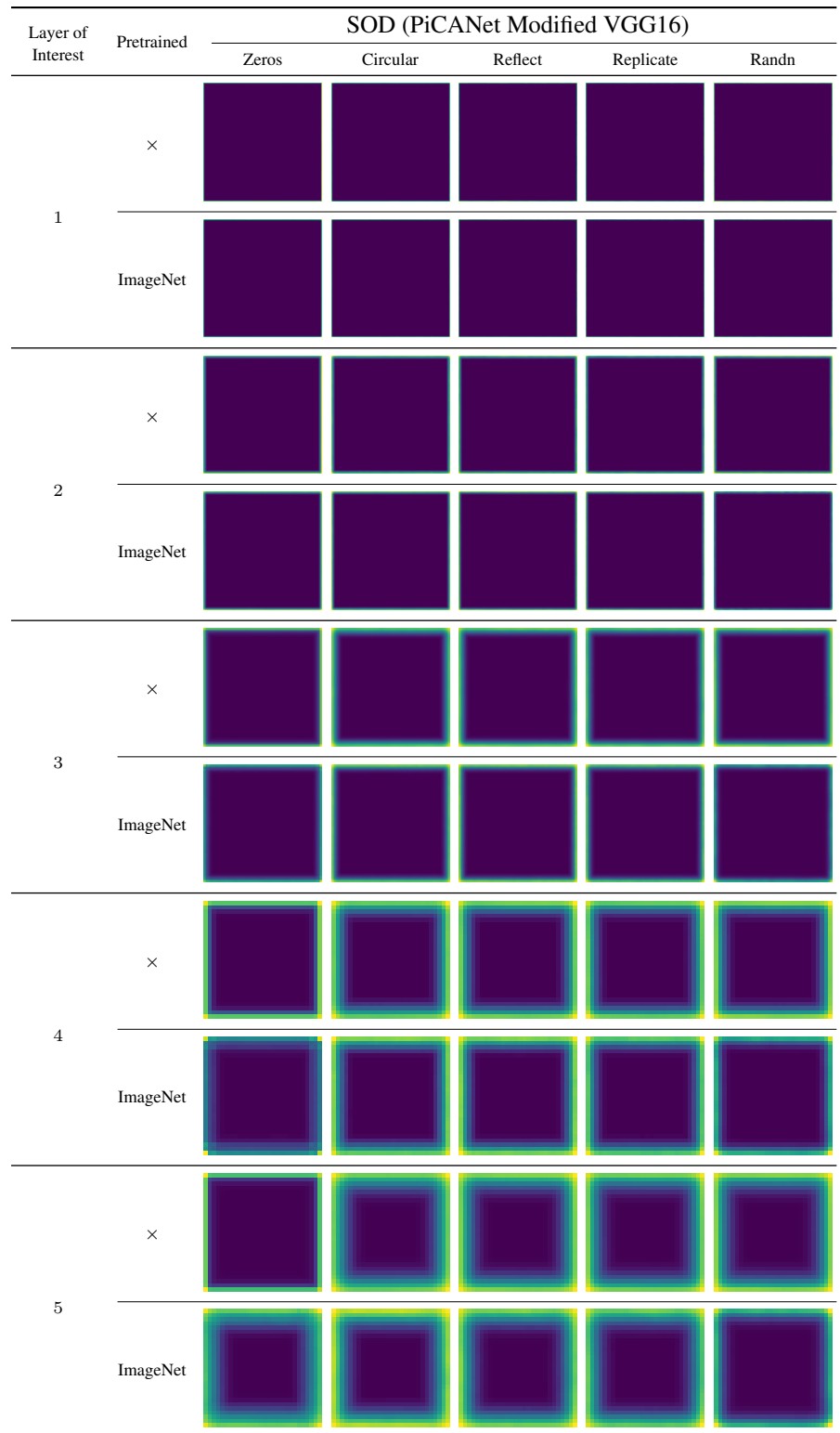

Figure 4: **Visualization of Position-Information Pattern from Padding (PPP).** The visualizations are calculated based on Eq. 3 over $480$ GMap samples. The results show that the pretrained model significantly reinforces PPP compared to randomly initialized networks. Note that each image is normalized to $[0, 1]$ separately, therefore the colors between images are not comparable.

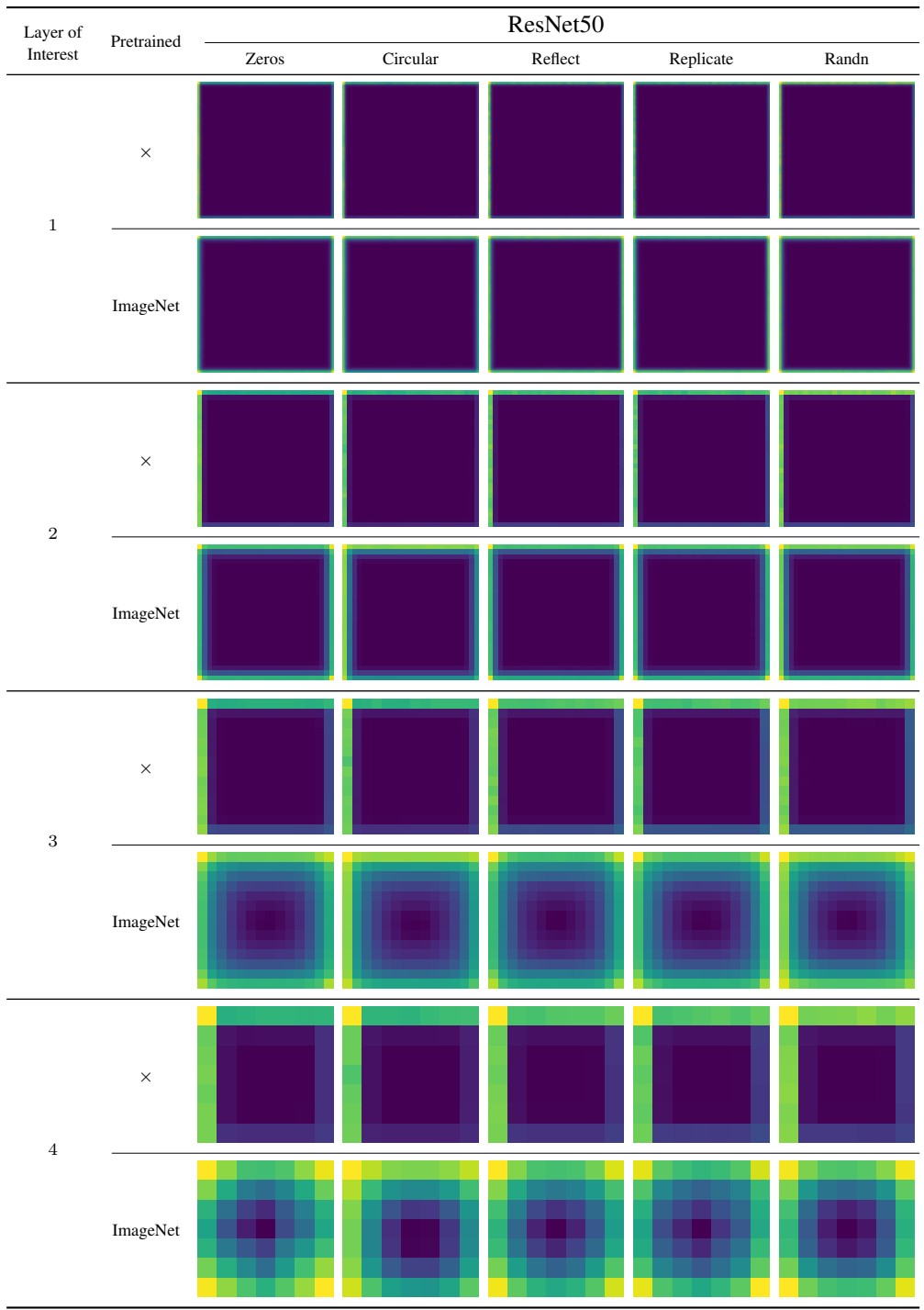

Figure 5: **Visualization of Position-Information Pattern from Padding (PPP).** The visualizations are calculated based on Eq. 3 over $480$ GMap samples. The results show that the pretrained model significantly reinforces PPP compared to randomly initialized networks. Note that each image is normalized to $[0, 1]$ separately, therefore the colors between images are not comparable.

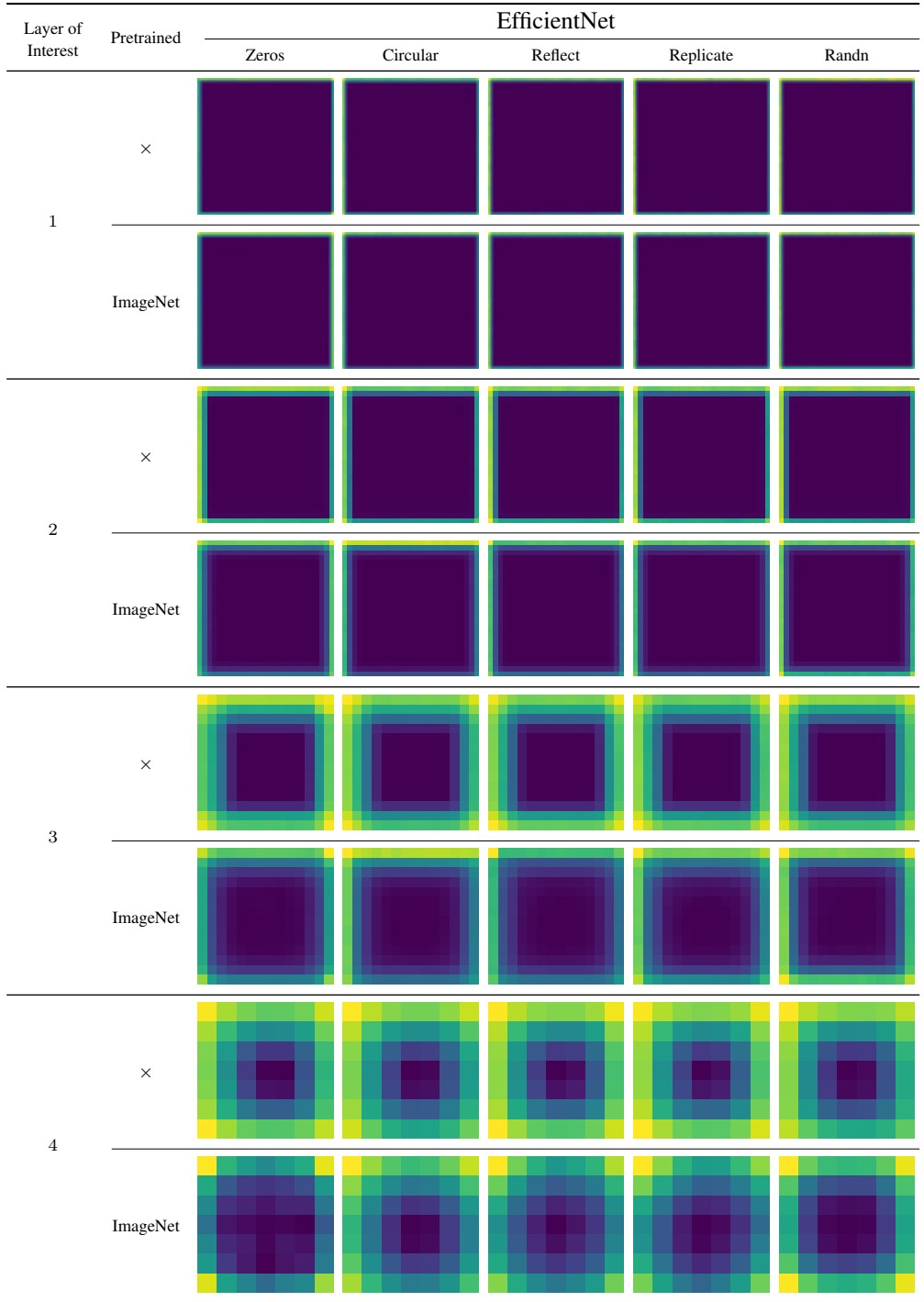

Figure 6: **Visualization of Position-Information Pattern from Padding (PPP).** The visualizations are calculated based on Eq. 3 over $480$ GMap samples. The results show that the pretrained model significantly reinforces PPP compared to randomly initialized networks. Note that each image is normalized to $[0, 1]$ separately, therefore the colors between images are not comparable.