# OpenReview forum: "Unveiling The Mask of Position-Information Pattern Through the Mist of Image Features"
_NeurIPS.cc/2022/Conference — NeurIPS 2022 Submitted_

### Official Review · Reviewer_MMJm · 2022-07-09

**Rating:** 4
**Confidence:** 4
**Soundness:** 2 fair
**Presentation:** 2 fair
**Contribution:** 2 fair

**Summary:**

This study investigates position information patterns learned due to the use of padding. A new evaluation method has been proposed by taking the difference between an optimal padding strategy and the “algorithmically-padded features”. The difference pattern becomes more recognizable when more samples are applied. Then signal-to-noise ratio and mean absolute error are proposed to measure the layer-wise difference between the two paddings. This whole system is claimed to be a better measure for the position information.


**Questions:**

The two metrics SNR-PPP and MAE-PPP, proposed in this paper are questionable. The use of signal-to-noise ratio and mean absolute error is not new, the new metric is essentially the proposed “optimal” padding, which can be used to compare against other paddings and compute differences, including SNR and MAE or any other metrics. Then let’s focus on the basic question, what the “optimal” padding is.

This proposed position measure is conducted at feature level, which is more difficult to draw conclusions than those studies at output level. Accuracy on the BHV test from [1] can be considered a measure against human perception, which is a golden standard in CNN understanding, e.g., why the classifier behaves differently than human and what could be the factor behind (absolute position). Similarly, the artificial ground-truth used in [2] can validate CNNs in a human readable way, if the model can output the wanted pattern based on position. This type of validation has also been applied in other CNN understanding papers[3,4].

The measure proposed in this paper is computed at the feature level, which is not human readable, or is still under exploration. Then the defined “optimal” padding becomes a golden standard to compare, which is very questionable. The paper also agrees with the “optimal” padding shown in Eqs. (1) and (2) “In practice, such an optimal-padding scheme is difficult to achieve.” I personally doubted if such padding exists in practice. Without further explanation, Sec2.2(Fig 1., line 81) directly shows the visual difference between the “optimal” and the “algorithmical” padding, this is really important but looks confusing. I can only guess the “optimal” refers to the randn padding from sec 2.4(please correct me if I am wrong), and the large difference to zero-padding is not surprising. The padded value by zero-padding is very different from other content-based, e.g., circular, reflect, replicate, which has been already studied in previous studies[1,5]. (In addition, the finding in Fig 3 and Table 1 that background color matters can already be found in previous study[5].) More importantly, I am still seeking an explanation why this randn padding could be the “optimal” to compute the two metrics, it looks more like another well-designed “algorithmical” padding. The “optimal” golden standard needs to be set and studied thoroughly, then the resulting findings (Table 2 and 3) can be analyzed reliably. Otherwise, it makes less sense no matter how high or low values a metric can achieve.

Therefore, I suggest a more thorough study on how the “optimal” padding can be achieved before computing metrics on top. And based on Table 1, the proposed randn padding behaves similarly to the replicate padding, which can be considered a special case drawn from the same distribution as randn padding.

Based on the questions above, I would suggest this study can be further improved for a better presentation and less confusion, the use of SNR and MAE is not main focus. More importantly, the design of "optimal" padding should be better addressed.

[1] Kayhan, Osman Semih, and Jan C. van Gemert. "On translation invariance in cnns: Convolutional layers can exploit absolute spatial location." Proceedings of the IEEE/CVF Conference on Computer Vision and Pattern Recognition. 2020.
[2] Islam, Md Amirul, Sen Jia, and Neil DB Bruce. "How much position information do convolutional neural networks encode?." arXiv preprint arXiv:2001.08248 (2020).
[3]Geirhos, Robert, et al. "ImageNet-trained CNNs are biased towards texture; increasing shape bias improves accuracy and robustness." arXiv preprint arXiv:1811.12231 (2018).
[4]Geirhos, Robert, Kristof Meding, and Felix A. Wichmann. "Beyond accuracy: quantifying trial-by-trial behaviour of CNNs and humans by measuring error consistency." Advances in Neural Information Processing Systems 33 (2020): 13890-13902.
[5]Islam, Md Amirul, et al. "Position, padding and predictions: A deeper look at position information in cnns." arXiv preprint arXiv:2101.12322 (2021).

**Ethics Review Area:**

["I don’t know"]

**Limitations:**

The paper can be further polished and re-organized to reduce misleading. The applicability of randn can be further discussed, the previously studied findings can be omitted.

**Strengths And Weaknesses:**

Strengths: The proposed idea of randn padding is plausible, a window is used to compute the mean and deviation values within. Then we can sample padding values based on the learned normal distribution. Addressing the effect of strided convolution also makes the investigation more comprehensive. The proposed randn padding can deliver less position information than other padding strategies.
Weakness: This study also has some essential problems that need to be addressed, please see the questions below.

---

> ### Author Response · Authors · 2022-07-28
> **Thanks for the review. Here are our responses.**
>
> We sincerely thank the comprehensive review. Here, we discuss the raised weaknesses and questions:
> 1. **`I can only guess the “optimal” refers to the randn padding.`**
>     - No, randn padding is also algorithmic padding as defined in Eq (1), which is different from optimal padding (defined in Eq (2)). We simulate the behavior of optimal padding with the satellite dataset introduced in Line 205. More specifically, the dataset is served as the S* set described in Line 64. And therefore all variables in Eq (1), (2), and (3) can be derived, as long as all images of S* are large enough.
>
>     - Here we describe one of the possible realizations of Eq (1), (2), and (3) below. Note that any implementations following the same definition will obtain the same results.
>
>          - Given a collection of satellite images at KxK pixels. Following Eq (1) and Eq (2), after a model is fully trained at NxN pixels, we can calculate the second term of Eq (3) (i.e., algorithmic padding part) by cropping NxN regions from the satellite images. Meanwhile, for the first term of Eq (3) (i.e., optimal padding part), we crop a much larger MxM region from the satellite images and set the model to no-padding mode. With such, the model automatically pads the features with realistic features produced by the pixels between the MxM region and the NxN region and follows the definition of optimal padding in Eq (2).
>
>         - Note that K can be any number larger than or equal to M (note M>N), and its value does not affect PPP as long as K>=M. The M value is architecture-dependent, and we obtain the values in Appendix A.
>
>     - We acknowledge that it may not be straightforward to derive such an implementation from the symbols in Eq (1), (2), and (3). We will add a section in Appendix to explicitly describe the aforementioned realization of the equations.
>
> 2. **`I suggest a more thorough study on how the “optimal” padding can be achieved.`** The procedure of obtaining optimal padding is described between lines 86-91. We describe one of the possible implementations above. Any approach following Eq (1), (2), and (3) will yield the same results.
>
> 3. **`SNR and MAE are not the main focus.`** Indeed, we did not emphasize these two functions as the core contribution, they are just a prefix of PPP. They are two simple functions aggregating the information reported by PPP, as PPP is a complicated and high-dimensional signal. Any reasonable aggregation (e.g., L2) can serve the purpose.

---

> > ### Author Response · Authors · 2022-08-06
> > **Please let us know whether you have additional quesitons**
> >
> >
> > Dear Reviewer MMJm,
> >
> > Thanks for the comments. We have replied to your questions and wonder whether you have additional comments or not. We would like to address all the questions that you have.
> >
> > Thanks,
> >
> > NeurIPS 2022 Conference Paper924 Authors

---

> > ### Comment · Reviewer_MMJm · 2022-08-08
> > **Feedback**
> >
> > Thank the authors for this explanation. However, this explanation makes the draft even harder to understand.
> > 1. Following the draft, from Section 2.1, Eqs(1) and (2) show the definitions of the "optimal" padding scenario(this is questionable). Then Section 2.2 discusses PPP, section 2.3 shows the "proposed" metrics, then Section 2.4 shows the "proposed" padding strategy. This organization is very misleading.
> >
> > The whole paper is built on top of the fundamental question for formulation Eqs(1) and (2), without a proper explanation or statement, the first proposed method is shown in Section 2.4, so naturally this would be considered related to the problem formulation Eq(2). Line 42-43, "To weaken the effect of PPP,.. Section 2.4", also misleads to this understanding.
> >
> > 2. I read lines 86-91 again, but it simply shows how to "calculate" the "optimal" padding, instead of explaining what the "optimal" padding should be. Line 86-91 and Eq(3) only show the PPP is defined as the "difference" between the "optimal" padding and an  algorithmical padding, no explanation on what the "optimal" is, as introduced in Eq(2). This question is the most important and fundamental to this study, if this this question is not clearly defined, then all the proposals are a little lame.
> >
> > Back to your explanation on the satellite images, there is no discussion or explanation about this in Section 2.1. More importantly, it is a little challenging to convince myself this cropped image can be considered the "optimal". If that was "optimal"(I personally doubt this), why wouldn't use that (or cropped images for vision tasks) but the proposed Randn?
> >
> > 3. The authors agree that they did not intend to emphasize the terms, but from the paper and the responses to other reviewers, MAE-PPP and SNR-PPP are still widely used to show their efficacy, this "contribution" can be simply interpreted as this paper applies MAE and SNR to measure the difference between the proposed "optimal" and an algorithmic padding, while why the proposed padding can mimic the "optimal" is not properly discussed and the definition of the "optimal" is omitted in this draft. Thus no matter what metrics are applied, (other than SNR or MAE), it can not solid measure the PPP.

---

> > > ### Author Response · Authors · 2022-08-08
> > > **Responses to the feedback**
> > >
> > > Thanks for responding to our rebuttal, we would like to further discuss some of the details as follows.
> > >
> > > 1. **`The definition of optimal padding`**
> > >     - To clarify, Eq (1) and Eq (2) are the definitions of paddings, following how images are captured from the physical world. It is not clear to us why the definition is questionable. Could the reviewer clarify which part of the definition is questionable or counter-factual? We would like to know what the reviewer means and address the issues, as the definition of PPP is entirely based on such a definition.
> > >     - Regarding the misunderstanding of randn padding is optimal padding, we will add an additional note to Section 2.4 that randn padding is algorithmic padding. Please note that Sections 2.1 to 2.3 have already given out a complete definition of optimal padding (Section 2.1), PPP (Section 2.2), and PPP metrics (Section 2.3). Section 2.4 does not have any argument or symbol that establishes it is optimal padding, or alternates any of the definitions that were already given in previous sections.
> > >
> > > 2. **`What the "optimal" is?`**
> > >     - Eq (2) has defined what the optimal padding should be, based on how images are captured from the physical world. It is the original pixels on S* without the S* -> S’ cropping (i.e.,  image capturing process). Since they are the ground truth pixels, therefore they are the optimal padding pixels.
> > >
> > >     - **`it is a little challenging to convince myself this cropped image can be considered the optimal`** The cropped image (S’) is not the source of optimal padding, the uncropped images (S*) define the optimal pixels (in Eq (2)). We use the satellite images to simulate the uncropped S* set, as they are much larger than the receptive field of the CNN models tested in the paper.
> > >
> > > 3. The terminology of using SNR-PPP and MAE-PPP
> > >     - Could the reviewer specify the alternative terms intended for us to switch to? The response seems to agree the terms MAE and SNR are not the main issues the reviewer intends to discuss. Instead, the main concern is still on the definition of “optimal” padding, which has no relation to the terminology of MAE-PPP and SNR-PPP. We would like to know what the reviewer means and address the issues.

---

### Official Review · Reviewer_rick · 2022-07-15

**Rating:** 3
**Confidence:** 3
**Soundness:** 3 good
**Presentation:** 3 good
**Contribution:** 2 fair

**Summary:**

Authors present a novel metric for detecting the presence and quantifying the strength of positional information encoding due to padding. They define PPP as a spatial statistic, i.e., expected absolute difference between algorithmically padded and optimally padded images’ activations, and summary statistics are defined from the spatial ones, i.e., SNR and MAE for PPP. In addition, they propose a new padding strategy, which aims to preserve local variability around a boundary when padding. The proposed metric is applied on pre-trained networks as well as those that are trained from scratch at different epochs. Results suggest that the position information encoding gets stronger with training, and very strong for pre-trained models. Compared to two existing alternatives, the proposed metric has lower variance, and therefore seems to be a more reliable metric.

**Questions:**

1.	My main suggestion is to better motivate the investigation and the developed metrics for applications of CNNs. How will this technology actually be used? A roadmap towards improving the current state of things using the PPP is unclear.
2.	What is the value of quantifying the strength of position encoding?  Better explanation towards this end would substantially improve the article.
3.	It is shown in figure 5, that proposed metrics are different at different layers. In this setting, it may be important for authors to provide some guidelines telling which layer one should look at. More specifically, difference (quantified with PPP) at which layer would cause the biggest problem for prediction accuracy?


**Limitations:**

Yes, they have.

**Strengths And Weaknesses:**

Strengths:
1.	The investigated concept is intriguing and potentially affect all application of CNNs.
2.	The proposed measure is very intuitive and easy to compute.
3.	Results suggest that the proposed metric shows lower variability compared to alternatives, and show that almost all networks encode position to some extent.
4.	Results showing emphasized position encoding for pre-trained models is interesting.

Weaknesses:
1.	A solution towards removing the position encoding is not discussed.
2.	Importance of quantifying the strength of PPP is not clear to me.
3.	Authors state that reliable PPP metrics are important for understanding PPP effects in different tasks. While this point is surely intriguing, such an explanation or understanding is not explicitly given in the article. Can the authors explicitly explain what type of understanding one reaches by looking at the PPP maps?
4.	The conclusion of the article remains a bit vague. While the proposed metrics have some more desirable attributes, value of these attributes for applications is unclear to me.  How will this actually improve the practice or our understanding?

---

> ### Author Response · Authors · 2022-07-28
> **Thanks for the review. Here are our responses.**
>
> We sincerely thank the comprehensive review. Here, we discuss the raised weaknesses and questions:
>
> 1. **`How will this technology actually be used?`**  The padding-related exploration is a fundamental building block to any existing or future architectures. Recently, researchers started to pay attention to paddings and the effect paddings induce. To better evaluate the efforts made in this field, we show that existing metrics are problematic, and we propose an improved metric. The proposed PPP metric serves as a better evaluation for any future methods related to positional information from padding. Furthermore, our results show that changing the padding scheme does not completely resolve the issue. The observation is a reminder to existing and future users adopting this workaround that the problem is not resolved yet.
>
> 2. **`A solution is not discussed.`**
>     - The major focus of the paper is to provide a reasonable metric to any future attempt toward understanding or removing positional information caused by paddings. To this end, the proposed PPP is the solution. A correct metric for the problem is required, especially when there are misunderstandings and erroneous conclusions drawn from previous publications in top-tier conferences (CVPR and ICLR). Without the correct understanding, improving the wrong metrics can never resolve the issue we are discussing. As shown in our paper, the proposed PPP metrics are improved replacements for previous metrics.
>
>
>     - As for the solution to the position information encoded by paddings, we believe the solution cannot be developed without reasonable metrics. Therefore, our whole paper aims at showing the problems of existing metrics, and demonstrating the new PPP metrics indeed resolve the issues.
>
> 3. **`Can the authors explicitly explain what type of understanding one reaches by looking at the PPP maps?`**
> Directly looking at the map only shows how strong PPP is, and how well our method can clearly extract it. SNR-PPP and MAE-PPP aggregate the PPP map into a score that can be used in comparing different models. We show and prove PPP is reliable (PPP is casted from the definition of padding, and shows a substantially lower deviation) compared to previous metrics. And discuss some observations in Section 4.
>
> 4. **`How will this actually improve the practice or our understanding?`**
>     - A reliable metric is always the first step to solving or understanding anything. Any information from an unreliable metric should be considered meaningless. Without any meaningful metrics, the severity of the problem cannot be correctly measured, and none of the solutions can claim it addresses the issue.
>
>     - Let’s say, the `accuracy` function in image classification has a 50% deviation without a reason (perhaps some ground truth classes are randomly swapped for some reason). Then it becomes obvious that we should fix the metric first, before starting benchmarking model performance on it.
>
> 5. **`What is the value of quantifying the strength of position encoding?`** First, a good metric should report a smooth and continuous value to express the property of interest. Second, if a problem cannot be analytically solved, a continuous value quantifies how well an alternative solution (e.g., a normalization or regularization) performs. To our understanding, the positional information problem is unlikely to be analytically resolved.
>
> 6. **`which layer one should look at`** As shown in Figure 5., PPP is progressively reinforced and becomes the strongest in the deepest layer (closest to the network prediction). The problem is caused by paddings throughout the network, therefore a genuine solution should reduce PPP in all layers.
>
> 7. **`at which layer would cause the biggest problem for prediction accuracy?`** In Figure 5., PPP is strongest in the final output layer, while being closest to the model prediction at the same time. Therefore, it is the last layer. But, again, PPP is not a problem of a particular layer, it is a problem caused by paddings throughout the whole network.

---

> > ### Author Response · Authors · 2022-08-06
> > **Please let us know whether you have additional quesitons**
> >
> > Dear reviewer,
> >
> > Thanks for the comments. We have replied to your questions and wonder whether you have additional comments or not. We would like to address all the questions that you have.
> >
> > Thanks,
> >
> > NeurIPS 2022 Conference Paper924 Authors

---

### Official Review · Reviewer_qWQu · 2022-08-02

**Rating:** 4
**Confidence:** 2
**Soundness:** 3 good
**Presentation:** 4 excellent
**Contribution:** 3 good

**Summary:**

This paper proposes a novel metric for measuring and visualizing the encoded positional information arising due to padding in Convolutional Neural Networks. They formally define the encoded information as PPP (Position-information Pattern from Padding) which measures the distance between feature maps arising from an algorithmic padding (such as zero padding) and an optimal notion of padding (i.e. the ground truth pixels that would be around the image). They demonstrate that PPP is a more reliable metric (low variance) than alternate metrics and hence it is useful for measuring the effectiveness of methods debiasing PPP due to padding. They show that PPP is a representation that the network develops as a part of its learning process.

**Questions:**

What can be more quantitative metrics to justify a good PPP metric? Currently, the majority of the focus is on the large standard deviation of PosENet. However, there must be other ways to compare which metric is a better representative of measuring the result of padding.

**Limitations:**

The authors already mention an important limitation of PPP being linked to the architecture of the network as it is evaluated on the outputs of a model.

**Strengths And Weaknesses:**

## Strengths
1. The paper is well-written, and contains sufficient details.
2. The proposed problem of determining the amount of position information encoded due to padding is important as it can be interfering to actual cues and degrade model performance.
3. The proposed metric is motivated well, and the analysis of PPP and prior works is thorough.
4. The idea of using an optimal padding scheme to quantify PPP is sound and reveals consistent patterns in experiments.
​
## Weaknesses
1. The evaluation is conducted on only one dataset of just 480 images. A more thorough investigation can be done by incorporating diverse datasets and observing the consistency of PPP as a metric over PosENet.
2. Fig. 5 and Section 4.3 demonstrate that PPP is a representation that the network favorably develops as a part of its learning process. Can the authors add some discussion on the implications of this? What new insights can be drawn from this experiment?

---

> ### Author Response · Authors · 2022-08-04
> **Thanks for the review. Here are our responses.**
>
> 1. **`Evaluate on only one dataset`**:
>     - For the number of samples, we want to highlight that our PPP metrics already achieve extremely low deviation with 480 images, therefore the number of samples is not an issue for the evaluation.
>     - We add two additional datasets to evaluate PPP metrics on all models presented in the paper. Finding a dataset with a large field-of-view with diverse content (as we want the optimal-padding features to be realistic) is non-trivial. Therefore, we collect 1024 images at 2048x2048 pixels, synthesized by InfinityGAN [1] trained on Flickr-Landscape and LSUN-tower. The evaluation results are attached with the following *anonymous* links:
>
>         - **Comparisons among three datasets**: https://i.imgur.com/hSv7YaW.png
>             - (red denotes the strongest, blue denotes the weakest)
>
>         - **Comparisons of randomly initialized networks**: https://i.imgur.com/hDtBlLf.png
>
>         The results show that (a) PosENet still maintains an unacceptable high deviation, (b) the significant PPP gain over model training is consistent with our previous observations, (c) MAE-PPP shows significantly better cross-dataset consistency than all other metrics, and (d) SNR-PPP sometimes recognizes certain models with high peak signal with relatively low MAE-PPP (which measures the average differences of PPP). A solution toward addressing PPP should suppress both SNR-PPP and MAE-PPP. We will include the additional results in the paper and add corresponding discussions.
>
>
> 2. **`Can the authors add some discussion on the implications of Fig. 5 and Section 4.3?`**
>     Thanks for the suggestion, we will add the following description to Section 4.3.
>
>     - “The results show that the network progressively learns to reinforce PPP as a favorable representation. Therefore, future methods in addressing PPP should be focusing on debiasing such a learning process (e.g., design paddings invisible to the network, or make the position-information less favorable with other replaced representations).“
>
>
> 3. **`What can be more quantitative metrics to justify a good PPP metric?`**
>     - For a metric, it is **essential** to have a low deviation. Otherwise, the numbers are not meaningful or comparable. PosENet is trivially dissatisfactory from this perspective, and shall not be used in any future studies.
>     - Could the reviewer provide more details on the expected other ways to compare the quality of a metric? From our perspective, the question implies a high deviation is not sufficient to prove PosENet is not a valid metric, where an additional metric is needed to address the issue. Some additional details on the issue can help us address the concern.
>
> [1] “InfinityGAN: Towards Infinite-Pixel Image Synthesis”, ICLR’22.

---

> ### Author Response · Authors · 2022-08-06
> **Please read our response and let us know if you have additional questions**
>
> Dear reviewer,
>
> Thanks for the comments. Please read our response to your questions and let us know if you have additional questions as the discussion phase ends on August 9.
>
> Thanks,
>
> NeurIPS 2022 Conference Paper924 Authors

---

### Meta-Review · Area_Chair_eti3 · 2022-08-26

**Recommendation:** Reject
**Confidence:** Certain

**Metareview:**

The three reviewers all leaned towards rejection for this paper. One reviewer was concerned with the relatively small number of images used in the experiment and how valid the conclusions can be from that for PPP as a better metric. Another confusion was over how optimality in padding can be defined. This was important because the MAE and SNR measures were based off of this.

**Award:**

No

---

### Decision · Program_Chairs · 2022-09-14

Reject